The relationship between steroid treatment and mortality in patients with COVID-19 followed up in an intensive care unit

Ozturk Huseyin Ali drozturkhuseyinali@gmail.com
Arici Fatih Necip
Department of Internal Medicine, University of Health Sciences—Adana Health Practice and Research Center , Adana , Turkey
Marunaka Yoshinori
Electronic publication date: 2025 Jan 17
Publication date: 2025
Volume: 13
Electronic Location ID: e18606
Received 2024 Aug 12; Accepted 2024 Nov 7
Copyright: ©2025 Ozturk and Arici
Copyright year: 2025
Copyright holder: Ozturk and Arici
License: This is an open access article distributed under the terms of the Creative Commons Attribution License, which permits unrestricted use, distribution, reproduction and adaptation in any medium and for any purpose provided that it is properly attributed. For attribution, the original author(s), title, publication source (PeerJ) and either DOI or URL of the article must be cited.
License URL: https://creativecommons.org/licenses/by/4.0/

Keywords: COVID-19, Mortality, Pulse methylprednisolone, Intensive care

Funding: The authors received no funding for this work.

==============================
Aim

Optimal treatment of the coronavirus disease (COVID-19) is still unclear. It has been reported that the use of different doses of corticosteroid treatments may reduce mortality. In our study, we aimed to find the effect of corticosteroid treatment dose on mortality of patients followed up in intensive care due to COVID-19.

Methods

Our retrospective, descriptive and single-centre study included 102 patients diagnosed with COVID-19 who were followed up in intensive care unit, 28 of whom received pulse steroids and 74 of whom received high dose steroids. Laboratory values, duration of intensive care unit and mortality rates of the patients were evaluated.

Results

Mortality was found to be statistically significantly lower in the group receiving pulse steroid compared to the group receiving high dose steroid. In multivariate logistic regression analysis, age and pulse steroid were found to be independent predictors of mortality. According to this analysis, each 10-year increase in age increased mortality by 4.8%, whereas pulse steroid decreased mortality by 79.4%.

Conclusion

In our study, we found that mortality was statistically significantly lower in the group of patients receiving pulse steroids than in the group receiving high dose steroids. We found that the number of patients using pulse steroids was statistically significantly lower in the group with mortality. We found that age and pulse steroid independently determined the patients with mortality.

Introduction

Coronavirus disease (COVID-19) is an infectious disease caused by a newly discovered coronavirus (SARS-CoV-2). The average incubation period is about 4 days and about 98% of those affected show symptoms within 11.5 days. Clinical manifestations may range from asymptomatic to pneumonia or acute respiratory distress syndrome (ARDS). This disease, which affects millions of people worldwide, is known to cause higher complications, mortality and morbidity, especially in older people and people with known hypertension, diabetes and obesity (Guan et al., in press; Sumbul et al., 2021; Koc et al., 2020). Disease severity can be clinically classified as mild, moderate, severe and critical (Wu & McGoogan, 2020). The treatment strategy of COVID-19 treatment with optimal efficacy and safety is still unclear. There are reports in the literature that a clinical picture associated with cytokine release syndrome in the course of COVID-19 disease can have serious and fatal consequences. Many authors have emphasised the importance of early recognition of the hyperinflammatory state resulting from a much more exaggerated and uncontrolled inflammatory response to SARS CoV-2. Due to the global spread of the pandemic, it has been reported that the use of different doses of corticosteroid treatments may reduce ARDS severity and mortality (Chaudhuri et al., 2021; Angus et al., 2020). Therefore, accurate and timely identification of the subgroup in which the type of corticosteroid treatment may reduce ARDS and mortality is important for planning an effective and safe treatment. In our study, we aimed to find the effect of corticosteroid treatment dose on mortality of patients followed up in intensive care due to COVID-19.

Materials and Method

Study population

Our study was designed as a retrospective, descriptive and single-center study. Between 01.05.2020–01.05.2022, patients who were diagnosed with COVID-19 at Adana City Training and Research Hospital, who received high-dose steroids or pulse steroids and had a history of adult intensive care unit hospitalisation were included. Polymerase chain reaction (PCR) positivity for SARS-CoV-2 and typical radiological findings (bilateral ground-glass areas) were the criteria for the diagnosis of COVID-19. All patients included in the study were moderately-serious patients who were hospitalised in intensive care units and needed oxygen support. Patients with similar age, gender and body mass index (BMI) were selected for the study groups. The study data were obtained from the data processing records of Adana City Training and Research Hospital and daily visit files of the patients. Detailed anamnesis and physical examination were performed in all patients; age, gender, medical history, physical examination findings and laboratory measurements were recorded in their files performed. Adana City Training and Research Hospital Ethics Committee approved the study with decision number 2784 dated 17.08.2023. Smoking history, whether the patients were receiving any oral or inhaled corticosteroid treatment were recorded. Lung tomographies of the patients were evaluated. Duration of intensive care unit stay, APACHE 2 scores, duration of mechanical ventilator stay, peep/fio2 (positive end-expiratory pressure/fraction of inspired oxygen) ratios and mortality rates were evaluated.

Corticosteroid treatment was categorised as pulse corticosteroid and high dose corticosteroid treatment. High dose corticosteroid treatment was defined as 1 mg/kg methylprednisolone or equivalent dose of dexamethasone daily. Pulse corticosteroid was defined as 500 mg methylprednisolone. The duration of pulse steroid treatment was 3 days. The study was conducted in accordance with the Declaration of Helsinki.

Laboratory measurements

Laboratory procedures of the study were performed in the Biochemistry Laboratory of Health Sciences University Adana City Training and Research Hospital. Laboratory parameters of the patients were measured with an automated chemistry analyser (Abbott Aeroset) using appropriate commercial kits (Abbott, Abbott Park, IL, USA).

Statistical analysis

All analyses were performed using the statistical software package SPSS 24.0 (Chicago, IL, USA). Student t-test or Mann–Whitney U was used for the comparison of continuous variables between groups. The chi-square (χ2) test was used to compare categorical variables. In the univariate analysis for the independent determination of mortality, multivariate logistic regression analysis was performed with statistically significant parameters with p value < 0.05. The sensitivity and specificity of the mortality predictive values were performed by ROC curve analysis. Statistical significance was accepted as p < 0.05 for all comparisons.

Table 1 Evaluation of the study groups in terms of demographic, clinical characteristics, laboratory values and intensive care follow-up.

Statistically significant results are in bold.

Variables	Group 1 n = 28	Group 2 n = 74	p	
Age (years)	61.3 ± 14.6	60.1 ± 14.8	0.711	
Gender, (M/F), n	20/8	45/29	0.312	
COPD, n	3 (10.7%)	5 (6.8%)	0.626	
CKD, n	6 (21.4%)	6 (8.1%)	0.168	
DM, n	12 (42.9%)	19 (25.7%)	0.185	
HT, n	15 (53.6%)	24 (32.4%)	0.133	
HF, n	4 (14.3%)	5 (6.8%)	0.332	
Pulse (pulse/minute)	86.6 ± 21.7	90.5 ± 17.8	0.367	
SBP (mmHg)	130.5 ± 23.4	127.4 ± 22.5	0.560	
DBP (mmHg)	75.5 ± 14.3	73.9 ± 12.6	0.595	
Fever (°C)	36.3 ± 0.41	36.3 ± 0.29	0.930	
BMI (kg/m2)	28.23 ± 3.50	28.1 ± 4.94	0.929	
Troponin (ng/l)	242.2 ± 792.2	60.7 ± 201.1	0.278	
Glucose (mg/dl)	182.7 ± 84.0	191.5 ± 93.9	0.664	
Urea (mg/dl)	78.5 ± 53.6	54.3 ± 38.0	0.035	
Creatinine (mg/dl)	1.96 ± 2.4	1.00 ± 0.75	0.052	
Sodium (mmol/l)	135.8 ± 4.94	137.2 ± 4.41	0.174	
Potassium (mmol/l)	4.27 ± 0.43	4.16 ± 0.62	0.424	
AST (U/L)	45.5 ± 24.0	44.0 ± 32.8	0.833	
ALT (U/L)	47.1 ± 27.4	43.4 ± 34.2	0.576	
LDH (U/L)	540.9 ± 190.9	491.8 ± 210.6	0.284	
Calcium (mg/dl)	8.32 ± 0.60	8.24 ± 0.60	0.537	
Ferritin (pg/ml)	854.9 ± 882.5	565.9 ± 760.1	0.105	
CRP (mg/L)	126.0 ± 88.7	130.5 ± 96.8	0.830	
WBC (10 ˆ3/pl)	12.7 ± 5.72	10.3 ± 6.49	0.094	
RBC (10 ˆ6/pl)	4.50 ± 0.69	4.28 ± 0.80	0.206	
Hemoglobin (g/dl)	13.0 ± 2.25	12.2 ± 2.07	0.109	
Neutrophil (10 ˆ3/pl)	11.18 ± 5.02	9.12 ± 6.01	0.110	
Lymphocyte (10 ˆ3/pl)	0.69 ± 0.54	0.64 ± 0.50	0.674	
INR, n	1.32 ± 1.76	1.02 ± 0.33	0.371	
D-dimer (pg/ml)	2449.2 ± 3087.7	1929.2 ± 2941.7	0.434	
Fibrinogen (mg/dl)	622.2 ± 138.7	604.6 ± 162.8	0.613	
Pro-BNP (pg/ml)	6023.4 ± 9905.7	2048.8 ± 3588.1	0.102	
Triglyceride (mg/dl)	201.7 ± 104.9	167.1 ± 78.0	0.246	
LDL (mg/dl)	100.2 ± 34.8	118.8 ± 35.3	0.199	
Days of intensive care hospitalisation, n	14.6 ± 10.1	15.8 ± 15.6	0.712	
PF Rate	159.0 ± 37.7	178.3 ± 46.2	0.052	
APACHE-2 score	21.1 ± 12.4	18.36 ± 10.31	0.269	
MV Days, n	4.92 ± 8.40	5.05 ± 12.0	0.959	
Mortality, n	8 (21.4%)	31 (41.9%)	0.041	
Notes.

COPD Chronic obstructive pulmonary disease

CKD Chronic kidney disease

DM Diabetes mellitus

HT Hypertension

HF Heart failure

SDB Systolic blood pressure

DBP Diastolic blood pressure

BMI Body mass index

AST Aspartate aminotransferase

ALT Alanine aminotransferase

LDH Lactate dehydrogenase

CRP C-reactive protein

WBC White blood cell

RBC Red blood cell

INR International Normalised Ratio

Pro-BNP Pro natriuretic peptide

LDL Low density lipoprotein

PF Rate Peep/Fio2 rate

APACHE-2 Acute Physiology and Chronic Health Evaluation

MV Mechanical ventilator

Table 2 Evaluation of the groups in terms of demographic, clinical characteristics, laboratory values and intensive care follow-up according to mortality.

Statistically significant results are in bold.

Variables	With mortality n = 37	Without mortality n = 65	p	
Age (years)	67.1 ± 13.8	56.7 ± 13.9	<0.01	
Gender, (M/F), n	18/19	19/46	0.058	
COPD, n	5 (13.5%)	3 (4.6%)	0.166	
CKD, n	4 (10.8%)	8 (12.3%)	0.768	
DM, n	13 (35.1%)	18 (27.7%)	0.464	
HT, n	18 (48.6%)	21 (32.3%)	0.121	
HF, n	5 (13.5%)	4 (6.2%)	0.275	
Pulse steroid use, n	6 (16.2%)	22 (33.8%)	0.042	
Pulse (pulse/minute)	91.1 ± 16.6	88.5 ± 20.2	0.509	
SBP (mmHg)	130.3 ± 24.0	127.0 ± 22.0	0.488	
DBP (mmHg)	71.1 ± 14.4	76.2 ± 14.06	0.53	
Fever (°C)	36.3 ± 0.42	36.3 ± 0.26	0.764	
BMI (kg/m2)	27.7 ± 5.34	28.4 ± 4.07	0.447	
Troponin (ng/l)	85.0 ± 268.4	118.3 ± 509.6	0.715	
Glucose (mg/dl)	188.9 ± 95.7	189.2 ± 88.9	0.988	
Urea (mg/dl)	62.5 ± 44.7	59.9 ± 43.8	0.761	
Creatinine (mg/dl)	1.17 ± 0.87	1.31 ± 1.74	0.638	
Sodium (mmol/l)	136.5 ± 5.38	137.0 ± 4.09	0.626	
Potassium (mmol/l)	4.22 ± 0.60	4.18 ± 0.56	0.712	
AST (U/L)	39.8 ± 27.0	47.0 ± 32.3	0.255	
ALT (U/L)	36.3 ± 21.8	49.0 ± 36.4	0.58	
LDH (U/L)	481.6 ± 184.1	518.8 ± 217.1	0.382	
Calcium (mg/dl)	8.3 ± 0.62	8.25 ± 0.58	0.692	
Ferritin (pg/ml)	701.7 ± 904.7	613.1 ± 742.1	0.594	
CRP (mg/L)	147.9 ± 84.8	118.7 ± 98.3	0.134	
WBC (103/pl)	11.2 ± 7.94	10.9 ± 5.3	0.809	
RBC (106/pl)	4.21 ± 0.63	4.42 ± 0.84	0.204	
Haemoglobin (g/dl)	12.1 ± 1.89	12.6 ± 2.26	0.236	
Neutrophils (103/pl)	9.9 ± 7.4	9.5 ± 4.7	0.811	
Lymphocytes (103/pl)	0.76 ± 0.60	0.59 ± 0.44	0.122	
INR, n	1.34 ± 0.97	0.97 ± 0.11	0.158	
D-dimer (pg/ml)	2139 ± 3514.0	2033.8 ± 2651.2	0.865	
Fibrinogen (mg/dl)	628.6 ± 140.8	598.4 ± 164.3	0.351	
Pro-BNP (pg/ml)	3027.2 ± 5650.5	2938.3 ± 6488.6	0.948	
Triglycerides (mg/dl)	197.4 ± 117.7	176 ± 77.5	0.499	
LDL (mg/dl)	119.7 ± 44.1	105.0 ± 32.2	0.367	
Days of intensive care hospitalisation, n	15.3 ± 7.43	15.5 ± 17.1	0.949	
PF Rate, n	165.6 ± 49.3	177.0 ± 41.7	0.221	
APACHE-2 score	23.3 ± 10.5	16.6 ± 10.5	0.004	
MV Days, n	5.91 ± 5.38	4.48 ± 13.4	0.537	
Notes.

COPD Chronic obstructive pulmonary disease

CKD Chronic kidney disease

DM Diabetes mellitus

HT Hypertension

HF Heart failure

SDB Systolic blood pressure

DBP Diastolic blood pressure

BMI Body mass index

AST Aspartate aminotransferase

ALT Alanine aminotransferase

LDH Lactate dehydrogenase

CRP C-reactive protein

WBC White blood cell

RBC Red blood cell

INR International Normalised Ratio

Pro-BNP Pro natriuretic peptide

LDL Low density lipoprotein

PF Rate Peep/Fio2 rate

APACHE-2 Acute Physiology and Chronic Health Evaluation

MV Mechanical ventilator

Table 3 Multivariate logistic regression analysis for the detection of mortality.

Statistically significant results are in bold.

Variables	Odd ratio	95% confidence interval	p	
Age (years)	1.048	1.008–1.089	0.019	
Pulse steroid use	0.206	0.062–0.678	0.009	
APACHE-2 Score	1.049	1.000–1.102	0.052	
Notes.

APACHE-2 Acute Physiology and Chronic Health Evaluation

Results

Patients with similar age and comorbidities were included in our study. Patients were divided into two groups as pulse steroid recipients (group 1) and high dose steroid recipients (group 2). Group 1 consisted of 28 patients, 20 males and eight females. In group 2, there were 74 patients, 45 males and 29 females. No statistically significant difference was found in terms of chronic obstructive pulmonary disease, diabetes, hypertension, heart failure, chronic kidney disease, pulse rate, systolic blood pressure, diastolic blood pressure, fever and body mass index. When the patient groups were evaluated in terms of biochemical parameters, urea level was found to be statistically significantly higher in group 1 than in group 2. No significant difference was observed when the groups were compared in terms of days of intensive care unit hospitalisation, APACHE-2 score and days on mechanical ventilator. When the groups were compared in terms of mortality, mortality rate was significantly lower in group 1 (Table 1).

The patients included in the study were divided into two groups according to mortality. Age was found to be statistically significantly higher and the number of patients using pulse steroid was found to be significantly lower in the group with mortality than in the group without mortality. When the groups were compared in terms of laboratory values, no significant difference was observed between both groups. When the groups were compared in terms of intensive care follow-up, APACHE-2 score was found to be statistically significantly higher in the group with mortality (Table 2).

In multivariate logistic regression analysis, age and pulse steroid use were found to be independent predictors of mortality. According to this analysis, each 10-year increase in age increased mortality by 4.8%, whereas pulse steroid use decreased mortality by 79.4% (Table 3).

ROC analysis was performed with the parameters found to be significant in multivariate logistic regression analysis. When the cut-off value of age was taken as 60.5, it was found to be 67.6% sensitive and 64.6% specific in determining mortality (Table 4, Fig. 1).

Table 4 Age-standardised roc analysis for the detection of mortality.

Variable	Area under the ROC curve	p	Cut-off	Sensitivity	Specificity	
Age	0.708 (0.603–0.813)	0.001	60.5	67.6%	64.6%	

Discussion

Acute Respiratory Distress Syndrome (ARDS) related to COVID-19 presents multifaceted challenges, stemming from hyper-inflammatory responses and underlying pathophysiological conditions. Our study aimed to investigate the efficacy of pulse versus high-dose corticosteroid treatments in influencing mortality rates among critically ill COVID-19 patients. The findings indicate that pulse steroid therapy is associated with a significantly lower mortality rate compared to high-dose steroid therapy, suggesting that treatment strategies tailored to mitigate inflammation may prove beneficial in this patient population. Previous research has established a robust link between systemic inflammation and ARDS progression, emphasizing corticosteroids’ role in attenuating this inflammatory response and improving outcomes (Meyer, Gattinoni & Calfee, 2021; Chang et al., 2022). The use of dexamethasone or other corticosteroids at equivalent doses is recommended for severe patients with COVID-19 receiving oxygen or respiratory support (RECOVERY Collaborative Group et al., 2021). The recommendation for corticosteroid use in severe COVID-19 cases aligns with our findings, reinforcing the theory that optimized steroid regimens can lead to improved survival rates (RECOVERY Collaborative Group et al., 2021; Chaudhuri et al., 2021). Specifically, the notable reduction in mortality associated with pulse steroid administration highlights the potential for lower-dose, intermittent steroid treatment to provide effective immunomodulation without the adverse effects often linked to continuous high-dose therapy. In our study, we found that mortality was statistically significantly lower in the group of patients receiving pulse steroids than in the group receiving high dose steroids. We found that age and pulse steroid use independently determined the patients with mortality.

In the study by Choi et al. (2023) patient groups with COVID-19 receiving pulse steroid and high dose steroid were compared. Mortality was found to be lower in the pulse steroid group. In addition, the duration of hospitalisation was shorter in the pulse steroid group. In the study by Okano et al. (2022) pulse steroid and high dose steroid groups were compared. In-hospital mortality and 28-day mortality were found to be lower in the group receiving pulse steroid. In another study by Okano, Sakurai & Yamazaki (2023), pulse steroid treatment was found to reduce 28-day mortality in intensive care unit patients compared to high-dose steroid therapy. In the study by Pinzón et al. (2021), it was found that recovery time and mortality were lower in the group receiving pulse steroid in intensive care unit compared to the group receiving high dose steroid. In our study, we found that mortality was lower in the group receiving pulse steroid. The study groups were similar in terms of age, comorbidities and laboratory values. The peep/fio2 ratio and APACHE-2 scores, which are parameters related to oxygen demand, were similar between both groups. For these reasons, it is important that we found lower mortality in the group receiving pulse steroid. In addition, while the group with high mortality and the group with low mortality were similar in terms of comorbidities and laboratory values, the low number of pulse steroid users in the group with high mortality is important. In multivariate logistic regression analysis, we found that pulse steroid use reduced mortality by 79.4%. Considering these, pulse steroid treatment may be preferred in appropriate patient groups.

Figure 1 ROC curve of age identifying patients with mortality in patients with Covid-19.

It is essential to acknowledge certain limitations in our study, including the small sample size compared to larger published trials, which may influence the generalizability of our findings. Additionally, the predominant use of methylprednisolone in our cohort, as opposed to dexamethasone employed in other studies, could contribute to variances observed in mortality rates and other outcomes. In the study of Batirel et al. (2021) patients were divided into 3 groups as standard care treatment, high dose steroid treatment and pulse-steroid treatment. In the results of the study, shorter intensive care unit stay was observed in the pulse steroid group. Mortality was similar between the two groups receiving steroid treatment (Batirel et al., 2021). In the study by Gundogdu et al. (2021) patients were divided into three groups as control group, high dose and pulse steroid treatment group. As a result of the study, it was found that pulse steroid treatment did not shorten the duration of hospital stay and did not reduce the need for intubation. In addition, similar mortality rates were found in both groups receiving steroid treatment (Gundogdu et al., 2021). In the study by Corral-Gudino et al. (2023), no significant difference was found between the group receiving pulse steroids and the group receiving high dose steroids in terms of mortality and duration of intensive care unit stay. In the study by Watanabe et al. (2023), no significant difference was found between the pulse steroid group and the high dose steroid group in terms of mortality and intensive care unit admission in patients hospitalised in intensive care unit. In our study, no significant difference was observed between the groups in terms of duration of intensive care unit stay. We included patients who received methylprednisolone 80 mg as high dose. In these studies, patients in the high dose group received dexamethasone 6 mg. This may be the reason why mortality was similar in these studies. In addition, this may have been due to the fact that the number of our patients was smaller compared to these studies. Importantly, the lack of statistically significant differences in ICU stay duration across treatment groups suggests that while pulse steroids may effectively address inflammation and reduce mortality, they do not necessarily expedite the recovery process in terms of hospital liberation. This finding highlights a critical area for further research, as understanding the interplay between inflammation control and recuperation time will be key to optimizing patient management.

Our study has some important limitations. We do not clearly know the duration of exposure to COVID-19 infection. New studies considering the duration of exposure to infection are needed. In our study, we do not have a control group that who did not receive steroid treatment. We included patients with moderate and severe COVID-19. New studies including the control group and patients with mild COVID-19 can be planned. New multicentre studies with a larger number of patients are needed.

Conclusion

In conclusion, our study adds to the growing body of evidence that pulse steroid treatment may serve as a more effective strategy in reducing mortality among critically ill COVID-19 patients, compared to high-dose corticosteroid treatments. Future randomized controlled trials are warranted to solidify these findings and optimize treatment protocols that address the complexities of ARDS in the context of COVID-19. Exploring patient characteristics that predict responsiveness to pulse steroids versus high-dose regimens may further enhance individualized patient care, ultimately improving survival outcomes in this vulnerable population.

Supplemental Information

Supplemental Information 1 Data set of study groups

Supplemental Information 2 Strobe checklist

Additional Information and Declarations

Competing Interests

Author Contributions

Human Ethics

Data Availability

The authors declare there are no competing interests.

Huseyin Ali Ozturk conceived and designed the experiments, performed the experiments, analyzed the data, prepared figures and/or tables, authored or reviewed drafts of the article, and approved the final draft.

Fatih Necip Arici conceived and designed the experiments, performed the experiments, analyzed the data, prepared figures and/or tables, authored or reviewed drafts of the article, and approved the final draft.

The following information was supplied relating to ethical approvals (i.e., approving body and any reference numbers): Adana City Training and Research Hospital Ethics Committee approved the study with decision number 2784 dated 17.08.2023.

The following information was supplied regarding data availability:

The raw measurements are available in the Supplementary File.

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
