# Peer review of "The relationship between steroid treatment and mortality in patients with COVID-19 followed up in an intensive care unit"

_PeerJ, doi:10.7717/peerj.18606_

## Round 0.1 · original submission · Major Revisions

Please revise your manuscript according to the reviewers' comments.
Yours,
Yoshi
Prof. Yoshinori Marunaka, M.D., Ph.D.

Reviewer 1 ·

Basic reporting

Despite your efforts, this article is poorly written.
The discussion is poorly written. Discuss your results and then discuss with the other studies.
English writing needs editing.
Abbreviations should be based on the text. The second aberration has no meaning

Experimental design

In cross-sectional studies, you cannot exclude information or disease unless you state it in the inclusion and exclusion criteria.

Validity of the findings

In my opinion, Table 1 has no meaning in cross-sectional studies.
You should describe the two study groups regarding demographic information, laboratory data, etc.

Annotated reviews are not available for download in order to protect the identity of reviewers who chose to remain anonymous.

Reviewer 2 ·

Basic reporting

The topic is interesting and quite well written. I have some suggestions.

Experimental design

1) Abstract. Results: Mortality was found to be statistically signiûcantly lower in the group
receiving pulse steroid compared to the group receiving high dose steroid. Multivariate
logistic regression analysis showed that pulse steroid use reduced mortality by 79.4%. I suggest to insert the most important staistically significant values to support the results.
2) Abstract. Conclusion: Pulse steroid treatment may reduce intensive care unit mortality in patients
with COVID-19. Abstract might be beneficial to include a sentence that briefly summarizes the key findings of the study. This can provide readers with a quick overview of the research.
3) Statistical analysis
108 All analyses were performed using the statistical software package SPSS 24.0 (Chicago,
109 IL, USA). The Kolmogorov-Smirnov test was used to evaluate whether the distribution of
110 continuous variables was normal. Continuous variables in group data were expressed as
111 mean±standard deviation. Categorical variables were expressed as numbers and percentages.
112 Student t-test or Mann-Whitney U was used for the comparison of continuous variables between
113 groups. I suggest to improve this part and underline the statistical tests used to evaluate the data.
4) Discussion:
145 ARDS is a catastrophic disease of multifactorial aetiology characterised by diffuse, severe
146 lung inflammation leading to acute hypoxaemic respiratory failure requiring mechanical
147 ventilation. There is a strong association between dysregulated systemic and pulmonary
148 inflammation and progression or delayed resolution of ARDS (7). Studies have shown that
149 systemic and pulmonary inflammation can regress with corticosteroid treatment and mortality
150 can be reduced (8). The discussion section needs to be improved. It is necessary to be more concise in the presentation of the facts, clarifying the results obtained and comparing them with previous or similar studies. However, it is interesting to answer the questions that arise from these results, backed up by published literature. Furthermore, underline the limitations of the study.
5) Conclusions. Underline the novelty of the study

Validity of the findings

The topic is interesting. Unfortunately, the sample is small and this affects the conclusions and the work. I suggets to comment this limitation during the revision.

Additional comments

None

---

## Round 0.2 · accepted · Accept

Congratulations!

Yours,

Yoshi
Prof. Yoshinori Marunaka, M.D., Ph.D.

Reviewer 2 ·

Basic reporting

no comment

Experimental design

no comment

Validity of the findings

no comment

Additional comments

no comment